# Monophasic Variant of *Salmonella* Typhimurium 4,[5],12:i:- (ACSSuGmTmpSxt Type) Outbreak in Central Italy Linked to the Consumption of a Roasted Pork Product (Porchetta)

**DOI:** 10.3390/microorganisms11102567

**Published:** 2023-10-15

**Authors:** Maira Napoleoni, Laura Villa, Lisa Barco, Claudia Lucarelli, Alessia Tiengo, Giulia Baggio, Anna Maria Dionisi, Antonio Angellotti, Ezio Ferretti, Simonetta Ruggeri, Monica Staffolani, Elena Rocchegiani, Valentina Silenzi, Benedetto Morandi, Giuliana Blasi

**Affiliations:** 1Centro di Riferimento Regionale Patogeni Enterici Marche, Istituto Zooprofilattico Sperimentale dell’Umbria e delle Marche “Togo Rosati”, Via Maestri del Lavoro, 7, 62029 Tolentino, Macerata, Italy; m.staffolani@izsum.it (M.S.); e.rocchegiani@izsum.it (E.R.); v.silenzi@izsum.it (V.S.); g.blasi@izsum.it (G.B.); 2Dipartimento di Malattie Infettive, Istituto Superiore di Sanità, Viale Regina Elena, 299, 00161 Roma, Italy; laura.villa@iss.it (L.V.); claudia.lucarelli@iss.it (C.L.); annamaria.dionisi@iss.it (A.M.D.); 3Centro di Referenza Nazionale e Laboratorio di Referenza WOAH per le Salmonellosi, Istituto Zooprofilattico Sperimentale Delle Venezie, Viale dell’Università, 10, 35020 Legnaro, Padova, Italy; lbarco@izsvenezie.it (L.B.); atiengo@izsvenezie.it (A.T.); gbaggio@izsvenezie.it (G.B.); 4UOC Igiene degli Alimenti di Origine Animale, Azienda Sanitaria Territoriale Fermo—Marche, Via Zeppilli, 22A, 63900 Fermo, Italy; antonio.angellotti@sanita.marche.it (A.A.); ezio.ferretti@sanita.marche.it (E.F.); simonetta.ruggeri@sanita.marche.it (S.R.); 5Laboratorio di Diagnostica Animale, Istituto Zooprofilattico Sperimentale dell’Umbria e delle Marche “Togo Rosati”, Via Maestri del Lavoro, 7, 62029 Tolentino, Macerata, Italy; b.morandi@izsum.it

**Keywords:** the monophasic variant of *Salmonella* Typhimurium, serovar 4,[5],12:i:-, MVST, roast pork, porchetta, RTE food, foodborne outbreak

## Abstract

The monophasic variant of *S*. Typhimurium 4,[5],12:i:- (MVST) is the third most commonly reported *Salmonella* serovar involved in human infections (8.8%) in the EU and ranks after *S*. Enteritidis (54.6%) and *S*. Typhimurium (11.4%). In Italy, in contrast, the MVST has achieved peculiar epidemiological and ecological success which has allowed it to be, since 2011, the serovar most frequently isolated from humans. In the summer of 2022, a foodborne outbreak of the MVST involving 63 people occurred in the Marche Region (Central Italy). A common food exposure source among some human cases was a roasted, ready-to-eat (RTE) pork product, porchetta, which is a typical product of Central Italy. This paper describes the results of investigations conducted to clarify this outbreak. The porchetta was produced by a local manufacturing plant and distributed to at least two local retail stores, one of which was the retail outlet for the manufacturing plant. The MVST was isolated from surface samples collected at the porchetta manufacturing plant and at both local retail stores via bacterial analysis, and the porchetta sampled at one store contained the MVST. These data confirm this type of RTE pork product can be a source of *Salmonella* infection in humans.

## 1. Introduction

Salmonellosis is the second most commonly reported gastrointestinal infection in humans after campylobacteriosis and is a major cause of foodborne outbreaks in the EU/EEA. The past three decades have seen the rapid, worldwide emergence of a new *Salmonella* serovar, namely the monophasic variant of *S*. Typhimurium (MVST), with the antigenic formula 4,[5],12:i:- [1]. This serovar was first identified in chicken carcasses in Portugal in the 1980s [2] and has gradually become prevalent in the swine chain and therefore in pork products, especially in Europe and in the United States [3]. This link to swine, pork, and pork products caused infections with the MVST to become a global public health emergency with regard to human infections [4].

In recent years, the MVST has overtaken *S*. Typhimurium (ST) in some countries including Italy, where it ranked as the top serovar isolated from humans in 2021 (1125 reported cases of the MVST against 360 ST cases) [4] and from food samples in 2019 (206 strains of the MVST against 62 strains of ST) [5]. In Italy, the MVST has spread so successfully that is has overtaken *S*. Enteritidis (SE), which is still the serovar most frequently identified as responsible for human and veterinary infections in the majority of European countries, according to the EFSA-ECDC One Health Zoonoses report from 2021 [6].

This rapid diffusion suggests that the MVST has a competitive advantage over ST. For example, according to D’Incau et al. [7], when considering the association between the clinical signs of salmonellosis in pigs and the serovar, the clinical signs are more associated with ST than with the MVST, rendering the infections caused by the MVST more difficult to recognize and thus less likely to be controlled.

Heavy metal and antibiotic resistance might be further reasons explaining the spread of the MVST. The presence of heavy metal resistance genes (HMRGs) could favor the MVST, meaning it better escapes the metal-mediated antimicrobial response of human macrophages [8], which are able to poison bacteria in the phagosome with large amounts of copper and zinc [9]. According to Mastrorilli et al. [10], among 50 epidemiologically unrelated isolates of the MVST found in Italy from 2010 to 2016, the main shared genetic trait was the presence of HMTGs encoding efflux systems involved in silver and copper tolerance as a consequence of the usual practice, especially in swine husbandry, of supplementing swine feed with copper and zinc to promote animal growth [11].

The antibiotic resistance patterns in the MVST also represent a gain of function for the evolutionary success of this serovar and its clones [3]. The first antibiotic resistance pattern associated with swine and pork products dates from 1997, namely the so-called Spanish clone [12], which showed plasmid-mediated resistance toward seven antimicrobial drugs, ampicillin, chloramphenicol, gentamicin, streptomycin/spectinomycin, sulfonamides, tetracyclines, and trimethoprim (ACGSSuTTmp type), and it was ascribed to sequence type (ST) 19. The European clone, ascribed to ST34, is characterized by chromosomally encoded resistance to ampicillin, streptomycin, sulphonamides, and tetracycline (ASSuT type), and has, over the years, progressively overtaken the Spanish and the United States clones. This latter clone, on the contrary, rarely shows multidrug resistance (MDR) patterns [3].

According to data from the European database of sales of veterinary antimicrobial agents [13], in swine breeding, the most commonly used antibiotics are tetracyclines, penicillins, and sulfonamides. This evidence could explain the specific antibiotic resistance pattern in the clones circulating in Europe. However, in combination with the usage of antimicrobials in animal husbandry and human medicine, the versatility of plasmids may have largely contributed to the spread of antimicrobial resistance in *Salmonella* enterica [14].

An additional favorable factor for the spread of the MVST is the presence in the MVST genome of type II toxin–antitoxin (TA) cassettes involved in persistence phenomena, as demonstrated by Mastrorilli et al. [10].

Many outbreaks due to the MVST have been reported to date in Europe and in the United States, and most of them have been linked to RTE food products. For example, two outbreaks were linked to the consumption of salami and pork products in 2009 [15] and 2010 [16], respectively, in North Italy; three different outbreaks were linked to the same product, dried pork sausage, in France [17,18] and in Spain [19] in 2011 and again in France in 2020 [20]; one outbreak was associated with roast pork in Spain in 2016 [21]; and a multi-state outbreak was linked to pork products in Washington State in 2015 [22].

Although the MVST is especially associated with swine products [23], over the years, several outbreaks of the MVST in Europe have been linked to different sources, such as pets in Italy in 2021 [24], chocolate products in 12 countries belonging to the European Union’s European Economic Area and the UK in 2022 [25,26], and tomatoes in Sweden in 2019 [27].

In the present study, we describe an outbreak of the MVST that occurred in the period of July-September 2022 in the Marche region (Central Italy) and involved 63 people. Epidemiological investigations carried out by the Prevention Department of the Marche Sanitary Local Health Authority in the territorial areas involved identified an RTE roast pork product, namely porchetta, as the most probable source of the outbreak when some patients declared that they had consumed this product and had purchased the porchetta at the same retail stores.

In this outbreak report, we present the results of a microbiological investigation conducted at the two retail stores (RS(A) and RS(B)) and at a food-processing plant (FPP) in order to trace the source of the infection.

## 2. Materials and Methods

### 2.1. The Collection of Human, Food, and Environmental Strains and an Epidemiological Investigation

Between 14 July and 7 September, 2022, the Regional Reference Centre for Pathogenic Enterobacteria (CRRPE) of the Marche region (Central Italy) of the Istituto Zooprofilattico Sperimentale of Umbria and Marche, Peripheral Health Structure of Tolentino (IZSUM), received an unusual number (*n* = 102) of human strains of *Salmonella* submitted for serotyping from the regional hospital analysis laboratories and the private analysis laboratories participating in human surveillance (Enter-Net Italia) for the Marche region.

The epidemiological investigations for the salmonellosis cases were carried out by the Prevention Department of the Local Health Authority (LHA) and were shared with the CRRPE with respect for anonymity, accordingly, as part of an IZSUM research project funded by the Ministry of Health titled, “Implementation of an integrated system for the management of apparently sporadic cases of salmonellosis through the use of the latest generation molecular techniques”. The LHA arranged interviews with the case patients using a standard epidemiological questionnaire. The cases were described according to demographics (area of residence, age, and gender), the date of onset of symptoms, clinical illness (the type of symptoms, hospitalization, and the duration of illness), and food consumption over a three-day period before the onset of illness.

It was therefore possible to identify two retail stores, RS(A) and RS(B), both located in the province of Fermo, where some patients had purchased the same type of food a few days prior to the onset of symptoms. Moreover, the food-producing plant (FPP) for this commonly mentioned food, also located in Fermo province, was identified. The potential source of infection was identified as an RTE roasted pork product, porchetta, which is a typical product of Central Italy.

In the framework of the outbreak investigation, the LHA inspected RS(A), RS(B), and the FPP, performing controls on traceability, the labeling of the porchetta, and the hygiene conditions at the premises. Furthermore, at the FPP, the procedures for good manufacturing and hygiene practices and those based on a hazard analysis and critical control points (HACCP) principles were assessed. Furthermore, the LHA gained access to the FPP’s self-assessment to verify the compliance of the food business operator (FBOp) with the microbiological criteria established by Regulation (EC) No 2073/2005.

During the inspections, the LHA performed food (porchetta) and environmental sampling at RS(A) and RS(B) on 17 August and 23 August, respectively, and at the FPP on 24 August.

Environmental samples were collected aseptically from food contact surfaces (FCSs) and non-food contact surfaces (NFCSs). The sampling was performed using sterile sponges (Whirl-Pak Speci-Sponge Bags) which were pre-hydrated with Dey-Engley neutralizing buffer.

At RS(A), 10 environmental samples were collected (via the sponge swab method) from unsanitized surfaces, i.e., the anteroom access door handle, a porchetta knife, a porchetta-picking scoop, a Teflon chopping board used to support and cut porchetta, a Teflon cutting board from the butcher’s area, a Teflon worktop in the butcher’s area, a Teflon cutting board for the white meat counter, a scale keyboard from the fresh meat area keyboard, a fresh meat slicer, and a Teflon cutting board from behind the counter.

At RS(B), the LHA collected one sample of porchetta and five environmental samples (these latter samples were collected via the sponge swab method) from unsanitized surfaces, i.e., a wooden chopping board for porchetta, a porchetta knife, a porchetta spatula, a Teflon cutting board to the right of the wooden cutting board, and a steel table to the left of the wooden chopping board.

Finally, at the FPP, LHA staff collected one sample of porchetta and six environmental samples from unsanitized surfaces in the meat-cooking areas, i.e., a knife, a refrigerator grill, a refrigerator wall, a transporting board for cooked porchetta, a refrigerator bottom, and a refrigerator handle, and two samples from sanitized surfaces in the raw-meat-processing areas, i.e., a raw-porchetta-handling table and a knife.

### 2.2. Microbiological Analysis and the Serotyping of Salmonella Strains

Food and environmental samples were analyzed to detect *Salmonella* at the IZSUM Laboratory according to UNI EN ISO 6579-1:2020 [28].

All *Salmonella* strains from humans and the food-related and environmental samples were serotyped by the CRRPE according to ISO/TR 6579-3:2014 [29].

Phenotypically, the MVST strains were also tested using a multiplex PCR with specific oligonucleotides for the presence of the gene *fljb* (flagellar antigen H:2) [30,31].

### 2.3. Antimicrobial Susceptibility Testing

The antimicrobial susceptibility of each of the MVST strains was determined via the disk diffusion method, according to the Clinical and Laboratory Standards Institute guidelines (CLSI, 2023), using ampicillin (AMP, 10 µg), amoxicillin–clavulanic acid (AMC, 30 µg), cefotaxime (CTX, 30 µg), ceftazidime (CAZ, 30 µg), cefoxitin (FOX, 30 µg), chloramphenicol (C, 30 µg), gentamicin (CN, 10 µg), meropenem (MEM, 10 µg), tetracycline (TE, 30 µg), trimethoprim–sulfamethoxazole (SXT, 23.75/1.25 µg), trimethoprim (TMP, 5 µg), sulfisoxazole (ST, 300 µg), pefloxacin (PEF, 5 µg), and streptomycin (S 10, µg). The reference strain *Escherichia coli* ATCC 25922 was used for antimicrobial susceptibility. The CLSI interpretive criteria for the disk diffusion susceptibility testing of *Salmonella* (CLSI, 2023) were used [32].

### 2.4. Multiple Locus Variable Number Tandem Repeats Analysis (MLVA)

All the MVST strains were tested via a multiple-locus variable-number tandem repeat analysis (MLVA) carried out by the Department of Infectious Diseases of Istituto Superiore di Sanità and the National Reference Centre for Salmonellosis of Istituto Zooprofilattico Sperimentale delle Venezie for human and for food and environmental samples, respectively.

The MLVA was performed following the laboratory SOP by the ECDC [33], and the MLVA profiles were each reported as a string of five characters (STTR9-STTR5-STTR6-STTR10-STTR3) representing the number of repeats at the corresponding locus.

### 2.5. Whole Genome Sequencing (WGS)

Whole genome sequencing (WGS) was performed on a representative selection of 30 strains based on the MLVA results and epidemiological information.

The DNA of all the 30 *Salmonella* strains was extracted using a commercial column-based protocol (QIAamp DNA Mini, QIAGEN, Valencia, CA, USA), and purified gDNA was quantified using a Qubit 3.0 Fluorometer (Life Technologies, Carlsbad, CA, USA). Libraries for whole genome sequencing were prepared using a Nextera XT DNA sample preparation kit (Illumina, San Diego, CA, USA). High-throughput sequencing was performed using a MiSeq Reagent kit v3, resulting in paired-end reads that were 301 bp’s long.

For the analysis of the WGS data, an in-house pipeline was used which included steps for the trimming and quality control check of the reads (Fastp) [34]. A genome assembly of the paired-end reads was performed using Shovill (https://github.com/tseemann/shovill; accessed on 21 August 2023) with the default parameters.

#### 2.5.1. In Silico Multilocus Sequence Typing (MLST)

The multilocus sequence typing (MLST) scheme used to characterize the *Salmonella* strains was based on a sequence analysis of the following seven housekeeping genes: chorismate synthase (aroC), β sliding clamp (dnaN), uroporphyrinogen-III synthase (hemD), histidinol dehydrogenase (hisD), *N*^5^-carboxyaminoimidazole ribonucleotide mutase (purE), 2-oxoglutarate decarboxylase (sucA), and fused aspartate kinase/homoserine dehydrogenase 1 (thrA).

The seven genes of the MLST scheme (ST) were deducted in silico using the program mlst v2.23.0 (https://github.com/tseemann/mlst) (https://pubmlst.org/) (accessed on 21 August 2023).

#### 2.5.2. Core Genome MLST

For a cluster analysis of the strains, core genome MLST (cgMLST), was performed according to an INNUENDO scheme of 3255 target loci (https://zenodo.org/record/1323684; accessed on 21 August 2023), using the chewBBACA v 3.0.0 allele calling algorithm [35]. Using the software GrapeTree v.1.5.0 [36], a minimum spanning tree (MSTreeV2) showing the relationships among the strains in terms of allelic mismatches was created.

Strains presenting seven or fewer allelic differences were considered to belong to the same cgMLST cluster.

## 3. Results

From 14 July to 7 September 2022, as part of Enter-Net Italia surveillance, an increase in the number of clinical strains of *Salmonella* submitted to the CRRPE of the Marche region was observed. In fact, the total number of strains received in the considered period (*n* = 102) was double the number collected in the same period over the two previous years, 48 in 2021 and 55 in 2020, suggesting the onset of an outbreak of salmonellosis.

### 3.1. Clinical Strains

#### 3.1.1. Serotyping and PCR Analysis

Seventy-eight of 102 *Salmonella* strains were serotyped and confirmed via a PCR as the MVST. The remaining 24 strains were other serovars.

#### 3.1.2. Antimicrobial Susceptibility Testing

Fifty-seven of the seventy-eight strains of the MVST showed resistance to the following antibiotics: ampicillin (A), chloramphenicol (C), streptomycin (S), sulfisoxazole (Su), gentamicin (Gm), trimethoprim (Tmp), and trimethoprim–sulfamethoxazole (Sxt) (ACSSu+Gm+Tmp+Sxt, i.e., these 57 strains had one “cluster strain” antibiotic resistance type). The peculiar antibiotic resistance to Gm, which is unusual for the main MVST clone circulating in Italy, allowed us to make a first definition in this study of a cluster on the basis of phenotype. Therefore, at first, this cluster strain’s antibiotic-resistance type was used to define the control strain with respect to the outbreak investigation against which all other potentially related isolates were compared to.

Furthermore, seven strains of the MVST had different partial types of antibiotic resistance with respect to the type of cluster strain: three strains (ACSSu+Tmp+Sxt), one strain (ACSSu), one strain (ACSSu+Gm), one strain (ACSSu+Amc+Gm+Tmp+Sxt), and one strain (CSSu+Gm+Tmp+Sxt).

The remaining 14 strains showed the following antibiotic resistance types: ASSuT (*n* = 8), ASSu (*n* = 2), ASFox (*n* = 1), ASSuFox (*n* = 1), ASSuGm (*n* = 1), and ASSuTPef (*n* = 1) (Table 1).

#### 3.1.3. Multiple-Locus Variable-Number Tandem Repeats Analysis (MLVA)

Sixty-three of the sixty-four strains of the MVST strains (fifty-seven with the cluster-strain antibiotic resistance type and seven with antimicrobial resistance types close to the cluster-strain type) showed the same MLVA profile corresponding to 3-14-10-na-211, while the remaining strains showed a closely related MLVA profile corresponding to 3-13-10-na-211. The other 14 MVST strains had different MLVA profiles.

Therefore, a total of 57 isolates from cases of human salmonellosis were identified as belonging to the same cluster of the MVST (antibiotic resistance type ACSSu+Gm+Tmp+Sxt and MLVA 3-14-10-na-211 or 3-13-10-na-211). Additionally, seven isolates from human cases were defined as being closely related to this cluster of the MVST (Table 1).

Therefore, from this point, either the cluster-strain antibiotic resistance type ACSSu+Gm+Tmp+Sxt or the MLVA profile 3-14-10-na-211 were considered reference characteristics defining the control strain with respect to the outbreak investigation.

#### 3.1.4. Epidemiological Investigations and Inquires in Neighboring Regions

The geographical distribution of the 64 human salmonellosis cases encompassed the entire Marche region and, therefore, all five local health authorities.

Thirty-three out of the sixty-four cases were male, and the most affected age group was 5–14 years old, with 26 cases. The cases were first reported between 14 July and 7 September 2022 and were distributed as follows: 22 in July, 37 in August, and 5 in September. The peak number of cases was recorded between 21 and 27 July (Figure 1). Twenty-nine patients of the sixty-four cases were hospitalized; for ten people out of the remaining thirty-five, information about their hospitalization was not available.

In 10 of the 43 epidemiological investigations that were conducted, the consumption of a food, a roasted, RTE pork product, porchetta, was commonly reported, together with the names of the retail shops from which the porchetta had been purchased. In the remaining 33 epidemiological investigations, this information was not described.

This allowed us to determine that one brand of porchetta produced by one FPP and sold at two retail outlets could be of interest. A food safety inspection at the FPP revealed poor basic hygiene practices and poor maintenance of the facility, particularly in the cooking areas and with respect to the equipment used for the production of porchetta. Furthermore, inadequate procedures regarding the identification and management of critical control points (CCPs) were revealed, e.g., the cooking and rapid cooling of porchetta were not considered by the food business operator (FBO) to be CCPs but were just managed using good manufacturing practices. In addition, conflicting information about the actual production process compared to the process described in the self-monitoring manual in relation to the temperatures used to cook and blast chill the pork meat product was provided by the FBO during an interview.

Following the verification by the LHA of the FPP’s self-assessment, no positivity for *Salmonella* spp. for the ready-to-eat products was found; only carcass swabs carried out for compliance with hygiene process criteria at the slaughterhouse of the FPP were positive. No serotyping data were available for these isolates.

Based on the results of the food safety inspections, the LHA suspended porchetta production at the FPP beginning on 31 August 2022. Furthermore, an information notification for attention (https://webgate.ec.europa.eu/irasff/notification/view/; notification number: 568075; accessed on 2 September 2023) was sent to the Rapid Alert System for Food and Feed. In addition, the LHA ordered extraordinary cleaning, disinfection, and maintenance processes to be carried out at the FPP, along with an expert review of permanent procedures based on HACCP principles.

Following the isolation of the cluster strain of the MVST from environmental samples obtained at RS(A) and RS(B), the LHA also prescribed the extraordinary sanitation of all the surfaces (whether they tested positive or not) and of all the equipment in all food-related areas in these retail shops. The training of the workers at all three locations (the FPP, RS(A), and RS(B)) with respect to the management of cooked products in order to prevent further recontamination was also prescribed.

The traceback carried out by the LHA allowed us to identify the suppliers of the pork meat used by the FPP located in the Abruzzo and Umbria regions. To determine whether the cluster strain was identified from clinical or food samples in the same period in the Abruzzo and Umbria regions, the peculiar feature of the circulating clone (e.g., its antibiotic resistance type) was shared with the CRRPE of Perugia (Umbria) and the CRRPE of Teramo (Abruzzo).

During the period of interest, no strains of the MVST which were of human or animal/food origin and characterized by the ACSSu+Gm+Tmp+Sxt antibiotic resistance type were identified by the CRRPE of Abruzzo. In the period of interest in Umbria, three strains of the MVST were isolated from pig carcasses at a slaughterhouse that was a supplier of pork meat for the FPP, but these strains had MLVA profiles unrelated to the MVST cluster.

In the same period, two clinical strains of the MVST isolated from two different Umbrian hospitals showed the same antibiotic resistance pattern as the cluster clone. No MLVA data were available for these strains.

### 3.2. Food-Related and Environmental Strains

#### 3.2.1. Microbiological, Serotyping, and PCR Analyses

Six of the twenty-five food and environmental samples tested positive for *Salmonella*, and the isolated strains were serotyped as the MVST (*n* = 5) and *S*. Infantis (*n* = 1) (Table 2). All the MVST strains were confirmed via a multiplex PCR. More details are reported in Appendix A.

The five samples which were positive for the MVST were a Teflon chopping board for supporting and cutting porchetta (unsanitized surface) at RS(A); a wooden chopping board for porchetta, a porchetta knife (unsanitized surface), and a porchetta sample at RS(B), and a transporting board for cooked porchetta (unsanitized surface) at the FPP.

#### 3.2.2. Antimicrobial Susceptibility and MLVA

All five food/environmental strains of the MVST had the antibiotic resistance type ACSSu+Gm+Tmp+Sxt and the MLVA profile 3-14-10-na-211, corresponding to those of the human cluster strains (Table 2).

### 3.3. Whole Genome Sequencing Analysis of the Clinical, Food-Related, and Environmental Strains

For the 30 genomes analyzed (25 clinical strains out of 64 total belonging to the MVST cluster on the basis of the MLVA results and all five food/environmental strains of the MVST) (Table 3), we obtained sequence data in accordance with the quality-control thresholds recommended for *Salmonella* as Q30 > 70%, average coverage ≥ 30×, a de novo assembly seq. length between 4.3 and 5.3 Mb, and the number of contigs ≤ 300 [37].

Furthermore, the MLST analysis confirmed that all 30 genomes belonged to ST34.

#### Cluster Analysis

The cgMLST analysis showed that no significant allelic distance was highlighted between the clinical, food-related, and environmental strains except for one clinical strain (Figure 2).

The central core included strains (*n* = 16) sharing the same allelic profile isolated from patients who consumed porchetta (*n* = 9) or from patients for whom this information was not available (*n* = 5), from porchetta (*n* = 1), and from the transporting board for cooked porchetta (*n* = 1). They showed the ACSSu+Gm+Tmp+Sxt “cluster strain” antibiotic resistance type (*n* = 15) and the ACSSu+Amc+Gm+Tmp+Sxt (*n* = 1) antibiotic resistance type and 3-14-10-na-211 MLVA profile (*n* = 16) (Figure 3).

Distances ranging from one to five alleles from the main group were seen for thirteen isolates, including strains from patients for whom information about the consumption of porchetta was not available (*n* = 10), from the Teflon chopping board used for supporting and cutting porchetta (*n* = 1), from the wooden chopping board for porchetta (*n* = 1), and from the porchetta knife (*n* = 1), with the ACSSu+Gm+Tmp+Sxt “cluster strain” antibiotic resistance type (*n* = 8) and different partial types of antibiotic resistance with respect to the type of cluster strain (*n* = 5) and the 3-14-10-na-211 (*n* = 12) and and 3-13-10-na-211 (*n* = 1) MLVA profiles, (Figure 3).

Only one strain, 22-40671, which was previously defined as being closely related to this MVST cluster on the basis of the MLVA profile 3-14-10-na-211 and the partial type of antibiotic resistance with respect to the type of cluster strain, i.e., ACSSu, was shown to be 49 alleles distant from the main group; therefore, it was considered a different clone (Figure 3).

## 4. Discussion

Our study provided evidence of the MVST contamination of both a porchetta sample and environmental samples from equipment used for handling this typical Italian food. The contaminated porchetta was the source of a foodborne outbreak involving 63 people.

Throughout the 1990s, pork production increased worldwide, resulting in consequent rapid globalization, and through the years, consensus on the close link between the MVST and the swine food chain became stronger, supported by the increasing evidence of human infections traced back to swine and pork products, especially RTE products [38].

In Italy, pork products are widely consumed [16]. In addition to production at the national level, there are many local traditional products such as porchetta, which is typical of Central Italy. Porchetta is an RTE product made from boned, seasoned, and roasted swine carcass meat. Roasting lasts from five to eight hours, depending on the size of the animal (a maximum weight of one quintal), followed by blast chilling down to 4 °C, which is measured in the thermal center of the product.

This type of pork product has already been identified as a source of *Salmonella* infections [21,39]. It is a food at risk of potentially spreading *Salmonella* due to the heterogeneous composition of its ingredients, which determines the variable pH and water activity levels depending on the sampling site.

Based on the porchetta production process, *Salmonella* can occur in this cooked, RTE pork product via two main routes. The first route is related to the use of *Salmonella*-positive carcasses or *Salmonella*-positive dissected raw meat; in this case, the pathogen is a consequence of meat contamination during slaughter or during the further handling procedures of boning, seasoning, and assembly [40].

According to the EFSA-ECDC EU One Health Zoonoses Report 2021 [6], considering all process hygiene criteria (PHC) monitoring data obtained from pig carcasses collected at slaughterhouses after dressing but before chilling, according to Regulation (EC) 2073/2005 and as reported by 23 EU member states (MSs), the overall proportion of *Salmonella*-positive samples based on official controls was 1.7% (N = 24,802), and it was significantly higher than that based on the FBOps’ own checks (1.4%, N = 103,270).

Regarding the Italian situation and bearing in mind that Italy is an MS that reports data collected by both the competent authority (CA) and the FBOps, 174 of 5147 samples reported by the CA (3.4%) and 107 of 11,494 samples reported by FBOps (0.93%) tested positive for *Salmonella*. Moreover, considering *Salmonella* along the swine production chain [41], for primary production, the most common *Salmonella* serovars causing human infections at the national level (i.e., the MVST and ST) are strictly associated with swine sources. To date, however, no control programs aimed at reducing the prevalence of *Salmonella* in swine farms have been implemented in Italy.

Furthermore, one of the main risks concerning carcass contamination is the persistence of *Salmonella* strains in slaughter or processing environments. Subsequent improper cooking procedures, such as inadequate thermal treatments, might not be sufficient to kill all present bacteria and can thus lead to the proliferation of *Salmonella* in the final product.

The second route for porchetta contamination occurs after cooking, whereby *Salmonella* originates either from the environment, through contaminated equipment or surfaces, or from healthy human carriers. The absence of competitive bacteria in RTE, cooked products like porchetta is another factor favoring the growth of *Salmonella* in a contaminated product if it is stored in unsuitable conditions.

The isolation of the MVST cluster clone at the FPP from a sponge swab on a transporting board used for cooked porchetta allowed us to exclude the post-cooking contamination of the porchetta at the two retail stores. At these locations, a Teflon chopping board used for supporting and cutting porchetta (RS(A)) and a wooden chopping board also used for porchetta, a porchetta knife (unsanitized surface), and porchetta (RS(B)) harbored the MVST cluster clone.

Nonetheless, it was not possible to establish the ultimate origin of the contamination at the FPP, nor it was possible to clarify whether the cluster clone originally contaminated the swine meat at the slaughterhouse level (i.e., the level of the meat suppliers) or was initially an environmental contaminant of the final product at the production level.

The evidence that no strain of the MVST was isolated from the swine carcasses at the Abruzzo slaughterhouse and that strains of the MVST with MLVA allelic profiles unrelated to the MVST cluster were isolated from the swine carcasses at the Umbrian slaughterhouse, both of which were suppliers of pork meat to the FPP, allows us to hypothesize that it is more probable that either the environmental contamination of the pork associated with some weaknesses in the porchetta production process or the contamination of the final product at the FPP occurred. The lack of evidence of *Salmonella* detected on a raw porchetta-manipulating table cannot exclude the potential environmental contamination of the meat since the table surface was sanitized prior to sampling.

The environmental contamination of the final product at the FPP also appears probable due to the detection of *Salmonella* on the transporting board used for cooked porchetta on this premises. The post-cooking contamination of the porchetta at the FPP could be the result of the persistence of the MVST strain in the FPP environment as a consequence of equipment being shared between raw- and cooked-meat-processing areas and the lack of sanitation procedures.

Following the positivity for *Salmonella* of the sampled porchetta, the Local Health Authority forced the FPP to cease production. At that point, the batch of porchetta that tested positive was no longer on the market as it had been sold and potentially already consumed due to its shelf-life.

Production activity at the FPP resumed on 12 October, following a favorable inspection carried out by the competent authority and the verification of compliance with the instructions previously given, i.e., by then, separate equipment was being used in the raw- and cooked-meat-processing areas, and CCPs were properly implemented for cooking and blast-chilling management.

The peculiar antibiotic resistance to gentamicin of the outbreak strain, which is unusual for the main MVST clone circulating in Italy, allowed us to obtain a first definition of the outbreak cluster on a phenotypic basis, resulting in the timely initiation of the investigation. This evidence underlines the importance of testing antibiotic resistance susceptibility as a preliminary screening method of characterizing *Salmonella* strains.

The identification of the unique MLVA profile of the investigated isolates of the MVST established, at first, the epidemiological relationship between the human cases and porchetta consumption and, therefore, the porchetta as the probable source of human infection.

This MLVA profile (3-14-10-na-211) is infrequent in the Italian human database, Enter-Net Italia (there were eight strains of the MVST of human origin with this profile from 2019 to date) and in the food–veterinary database, Enter-Vet (28 isolates from 2019 to date: 8 in 2019, 3 in 2020, 3 in 2021 and 14 in 2022; the isolates were of swine origin, with the exception of the 2022 isolates, which were all of bovine origin and were transmitted from a single laboratory).

The cgMLST analysis performed on a selection of 30 isolates of the MVST confirmed the belonging of clinical (*n* = 24), food-related (porchetta) (*n* = 1), and environmental (*n* = 4) strains to the same cluster except for one clinical strain that was shown to be 49 alleles distant from the main group and was therefore considered a different clone. Regarding this strain, it was at first considered closely related to the MVST cluster on the basis of the partial type of antibiotic resistance (ACSSu) with respect to the type of cluster strain (ACSSu+Gm+Tmp+Sxt) and of the MLVA profile (3-14-10-na-211). The epidemiological investigation surrounding this case was not carried out due to a failure to report to the Prevention Department of the Marche Sanitary Local Health Authority. Therefore, it is plausible that the clinical case in question never bought nor consumed porchetta.

Regarding the remaining 40 clinical strains which were not analyzed via WGS, since they showed the cluster-strain antibiotic resistance type ACSSu+Gm+Tmp+Sxt and the MLVA cluster strain profile 3-14-10-na-211, we included them in the cluster which, therefore, involved 63 people in total.

Regarding the evidence that the 30 genomes were ascribed to ST34, this sequence type is typical of the European clone [42], which, has gradually overtaken the Spanish and U.S. clones to become the most prevalent clone worldwide at present [43]. However, the isolates belonging to ST34 show the typical ASSuT phenotype. The antibiotic resistance type of this cluster strain, ACSSu+Gm+Tmp+Sxt, could be the result of a selective pressure imposed by the use of antimicrobials in the farms of origin of the pork used later for the production of porchetta. Following the WGS evidence of the inclusiveness in the cluster of six out of seven strains showing partial types of antibiotic resistance with respect to the type of cluster strain, the resistance pattern ACSSu+Gm+Tmp+Sxt could also be explained by the European clone’s plasmid-mediated acquisition of antimicrobial resistance genes and their subsequent loss.

Further phylogenetic analyses of this clone will be needed to better understand its origin and its evolution inside the ST34 population. The presence of specific genes of antimicrobial and heavy metal resistance, as well genes against biocides that could have favored the persistence of this clone of the MVST in the FPP environment, will be also investigated.

## 5. Conclusions

According to the Food and Agriculture Organization of United Nations, global pork production is expected to increase by 13.1% by 2030 and account for 34% of global meat consumption [44].

If the routes through which *Salmonella* can enter the swine chain generally remain unclear [45], it is equally true that a structured control plan, as has been put in place for the poultry industry, could allow for the identification and therefore the control of *Salmonella* along the chain, with the consequent containment of its spread to humans via food.

The key factor in the management of this investigation was, indeed, the intersectoral collaboration among microbiologists, veterinarians, and health and food safety authorities at national, regional, and local levels.

The laboratory-based surveillance network and consolidated system for contact and collaboration, like the one implemented in Italy’s Marche region, was essential in detecting the outbreak in a timely manner and for conducting a further investigation to identify the source of the outbreak clone.

Therefore, the content of this manuscript may be especially relevant for food and public health laboratories and epidemiologists.

## Figures and Tables

**Figure 1 microorganisms-11-02567-f001:**
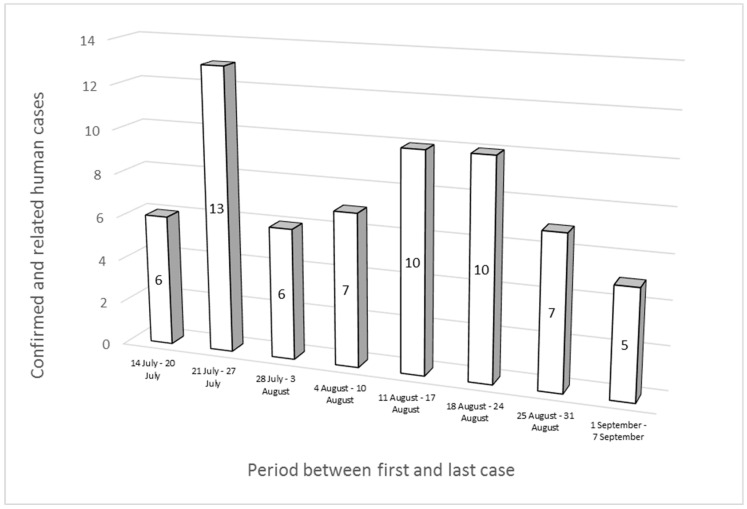
Epidemic curve of “MVST clone” cases (*n* = 57) and “MVST clone—related” cases (*n* = 7).

**Figure 2 microorganisms-11-02567-f002:**
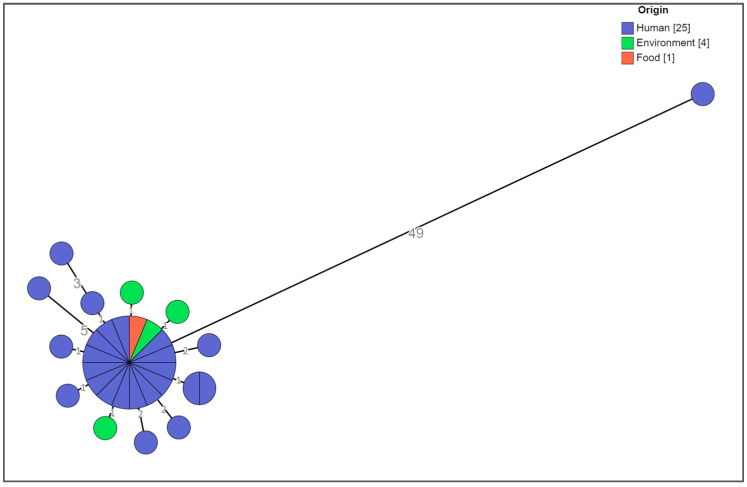
Cluster analysis of a selection of the MVST strains colored by origin. The numbers reported in the branches indicate the allelic differences existing between the strains.

**Figure 3 microorganisms-11-02567-f003:**
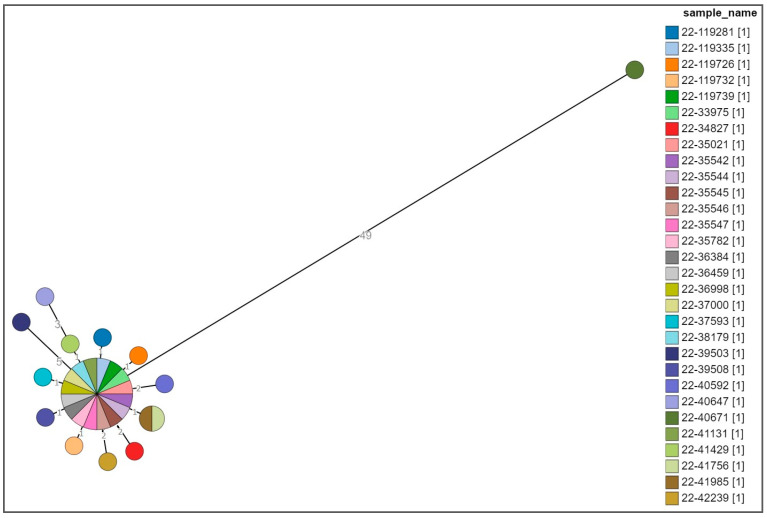
Cluster analysis of a selection of the MVST strains. Different colors are representative of the identification detail of each strain.

**Table 1 microorganisms-11-02567-t001:** Antimicrobial susceptibility testing, multiple-locus variable-number tandem repeat analysis (MLVA) typing, and inclusiveness in the cluster of human strains of the MVST (*n* = 78). The following are the criteria of inclusiveness: the “cluster strain” antibiotic resistance profile ACSSu+Gm+Tmp+Sxt and the MLVA profile 3-14-10-na-211; the “cluster strain” antibiotic resistance profile ACSSu+Gm+Tmp+Sxt and an MLVA profile different for just one tandem repeat in one locus compared to the MLVA profile 3-14-10-na-211; different partial types of antibiotic resistance with respect to the type of cluster strain (at least four antimicrobials shared) and the MLVA profile 3-14-10-na-211.

No. of Strains	Origin of Sample	Antibiotic Resistance Type *	MLVA Profile	Inclusiveness in the MVST Cluster
56	Feces (55) + Urine (1)	ACSSu+Gm+Tmp+Sxt	3-14-10-na-211	Yes
1	Feces	ACSSu+Gm+Tmp+Sxt	3-13-10-na-211	Yes
3	Feces	ACSSu+Tmp+Sxt	3-14-10-na-211	closely related
1	Feces	ACSSu	3-14-10-na-211	closely related
1	Feces	ACSSu+Gm	3-14-10-na-211	closely related
1	Feces	ACSSu+Amc+Gm+Tmp+Sxt	3-14-10-na-211	closely related
1	Feces	CSSu+Gm+Tmp+Sxt	3-14-10-na-211	closely related
3	Feces	ASSuT	3-13-10-na-211	No
1	Feces	ASSuT	2-11-10-na-211	No
1	Feces	ASSuT	3-11-10-na-211	No
1	Feces	ASSuT	3-11-11-na-211	No
1	Feces	ASSuT	3-12-11-na-211	No
1	Feces	ASSuT	3-12-17-na-211	No
1	Feces	ASFox	3-12-8-na-211	No
1	Feces	ASSu	3-12-14-na-211	No
1	Feces	ASSu	3-12-9-na-211	No
1	Feces	ASSuFox	3-12-16-na-211	No
1	Feces	ASSuGm	3-12-8-na-211	No
1	Feces	ASSuTPef	3-15-10-na-211	No

* Ampicillin (A), chloramphenicol (C), streptomycin (S), sulfisoxazole (Su), gentamicin (Gm), trimethoprim (Tmp), trimethoprim–sulfamethoxazole (Sxt), amoxicillin–clavulanic acid (Amc), tetracycline (T), cefoxatin (Fox), and pefloxacin (Pef).

**Table 2 microorganisms-11-02567-t002:** Antimicrobial susceptibility testing and the MLVA typing of food and environmental strains of the MVST (*n* = 5).

No. of Strains	Place of Sampling	Sample Detail	Origin of Sample	Antibiotic Resistance Type **	MLVA Profile
1	RS(A)	Sponge swab on an unsanitized surface	Teflon chopping board for supporting and cutting porchetta (FCS) *	ACSSu+Gm+Tmp+Sxt	3-14-10-na-211
1	RS(B)	Ready-to-eat food	Porchetta	ACSSu+Gm+Tmp+Sxt	3-14-10-na-211
1	RS(B)	Sponge swab on an unsanitized surface	Wooden chopping board for porchetta(FCS) *	ACSSu+Gm+Tmp+Sxt	3-14-10-na-211
1	RS(B)	Sponge swab on an unsanitized surface	Porchetta knife (FCS) *	ACSSu+Gm+Tmp+Sxt	3-14-10-na-211
1	FPP	Sponge swab on an unsanitized surface	Transporting board for cooked porchetta (FCS) *	ACSSu+Gm+Tmp+Sxt	3-14-10-na-211

* FCS: urface in contact with food. ** Ampicillin (A), chloramphenicol (C), streptomycin (S), sulfisoxazole (Su), gentamicin (Gm), trimethoprim (Tmp), and trimethoprim–sulfamethoxazole (Sxt).

**Table 3 microorganisms-11-02567-t003:** List of strains analyzed using WGS and epidemiological and analytic data.

ID Strain	Origin of Sample	Identification Detail	Consumption of Porchetta by Cases	Antibiotic Resistance Type	MLVA Profile
22-35542	Human	Feces	Confirmed	ACSSu+Gm+Tmp+Sxt	3-14-10-na-211
22-35544	Human	Feces	Confirmed	ACSSu+Gm+Tmp+Sxt	3-14-10-na-211
22-35546	Human	Feces	Confirmed	ACSSu+Gm+Tmp+Sxt	3-14-10-na-211
22-36384	Human	Feces	Confirmed	ACSSu+Gm+Tmp+Sxt	3-14-10-na-211
22-36459	Human	Feces	Confirmed	ACSSu+Gm+Tmp+Sxt	3-14-10-na-211
22-35782	Human	Feces	Confirmed	ACSSu+Gm+Tmp+Sxt	3-14-10-na-211
22-36998	Human	Feces	Confirmed	ACSSu+Gm+Tmp+Sxt	3-14-10-na-211
22-37000	Human	Feces	Confirmed	ACSSu+Gm+Tmp+Sxt	3-14-10-na-211
22-38179	Human	Feces	Confirmed	ACSSu+Gm+Tmp+Sxt	3-14-10-na-211
22-35021	Human	Feces	Information not available	ACSSu+Gm+Tmp+Sxt	3-14-10-na-211
22-35545	Human	Feces	Information not available	ACSSu+Gm+Tmp+Sxt	3-14-10-na-211
22-35547	Human	Feces	Information not available	ACSSu+Gm+Tmp+Sxt	3-14-10-na-211
22-34827	Human	Feces	Information not available	ACSSu+Gm+Tmp+Sxt	3-14-10-na-211
22-41131	Human	Feces	Information not available	ACSSu+Gm+Tmp+Sxt	3-14-10-na-211
22-41429	Human	Feces	Information not available	ACSSu+Gm+Tmp+Sxt	3-14-10-na-211
22-40647	Human	Urine	Information not available	ACSSu+Gm+Tmp+Sxt	3-14-10-na-211
22-41985	Human	Feces	Information not available	ACSSu+Gm+Tmp+Sxt	3-14-10-na-211
22-37593	Human	Feces	Information not available	ACSSu+Gm+Tmp+Sxt	3-13-10-na-211
22-40592	Human	Feces	Information not available	ACSSu+Tmp+Sxt	3-14-10-na-211
22-41756	Human	Feces	Information not available	ACSSu+Tmp+Sxt	3-14-10-na-211
22-42239	Human	Feces	Information not available	ACSSu+Tmp+Sxt	3-14-10-na-211
22-40671	Human	Feces	Information not available	ACSSu	3-14-10-na-211
22-39503	Human	Feces	Information not available	ACSSu+Gm	3-14-10-na-211
22-33975	Human	Feces	Information not available	ACSSu+Amc+Gm+Tmp+Sxt	3-14-10-na-211
22-39508	Human	Feces	Information not available	CSSu+Gm+Tmp+Sxt	3-14-10-na-211
22-119335	Food	Porchetta	//	ACSSu+Gm+Tmp+Sxt	3-14-10-na-211
22-119726	Environment	Teflon chopping board for supporting and cutting porchetta	//	ACSSu+Gm+Tmp+Sxt	3-14-10-na-211
22-119281	Environment	Wooden chopping board for porchetta	//	ACSSu+Gm+Tmp+Sxt	3-14-10-na-211
22-119732	Environment	Porchetta knife	//	ACSSu+Gm+Tmp+Sxt	3-14-10-na-211
22-119739	Environment	Transporting board for cooked porchetta	//	ACSSu+Gm+Tmp+Sxt	3-14-10-na-211

## Data Availability

The data presented in this study are available on request from the corresponding author. The data are not publicly available due to privacy.

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
