# Peer review of "Monophasic Variant of Salmonella Typhimurium 4,[5],12:i:- (ACSSuGmTmpSxt Type) Outbreak in Central Italy Linked to the Consumption of a Roasted Pork Product (Porchetta)"

_microorganisms, 2023, doi:10.3390/microorganisms11102567_

Round 1
Reviewer 1 Report (Previous Reviewer 1)
The new version of the manuscript has been greatly improved taking into account the comments I sent on the previous version.
I don't have any other comments except I'm wondering if the MVST studied isolates have been entered in the Enter-Net and Enter-Vet databases.
Author Response
The studied MVST isolates of clinical origin have been entered in the Enter-Net database, the food-related and environmental ones in Enter-Vet.
The entering of the epidemiological and analytical data of every single Salmonella strain serotyped at the Regional Reference Centre for Pathogenic Enterobacteria (CRRPE) of the Marche Region of IZSUM in the Enter-Net and Enter-Vet databases is an activity that is routinely carried out regardless of the occurrence of outbreak.
Reviewer 2 Report (Previous Reviewer 3)
The authors have addressed the majority of my concerns from my previous review. However, there are still a few errors in English usage and typos that should be fixed by the authors.
Another proof reading would make sure all the errors have been addressed.
Author Response
The manuscript has been revised by an English native speaker for a second time. Attacched below the revised version.

This manuscript is a resubmission of an earlier submission. The following is a list of the peer review reports and author responses from that submission.
Round 1
Reviewer 2 Report
The rational design and logic flow should be improved. The aim should be clearly clarified in the introduction, where the segregated points dilute the idea.
We need molecular test or strain subtyping assays for further outbreak tracking and investigation, however, this is not available in current study. I highly suggest whole genomic sequence, PFGE, or other typing method (10.1128/spectrum.02479-22) should be implemented to address the issue here.
The control strains regarding to this outbreak investigation should be added.
The quality of figure is low.
References in general are very old, and a few key work in the field should be acknowledged.
10.1128/spectrum.03119-22
10.3390/pathogens11121500
10.1099/mgen.0.000897
10.1016/j.amsu.2022.104597
10.2807/1560-7917.ES.2022.27.15.2200314
10.2807/1560-7917.ES.2019.24.47.1900643
10.2807/1560-7917.ES.2019.24.34.1900207
10.2807/1560-7917.ES.2018.23.13.17-00375
10.3390/microorganisms7090298
10.3389/fmicb.2019.00985
10.3201/eid2310.161248
a much fragmented introduction, the logical flow should be improved.
Reviewer 3 Report
This study investigates an outbreak of Salmonella Typhimurium, monophasic variant associated with porchetta. After an outbreak was detected by an increase in ill victims reporting to public health services, human isolates were serotyped and found to be STMV lacking he fljB gene by PCR. further investigations identified porchetta as the likely vehicle for the infections. Investigations of processing facilities detected STMV on various surfaces. The authors then speculate about the method of contamination including under cooking and cross contamination of cooked product with STMV.
The study is fairly well done, however, the manuscript suffers from poor English usage that needs to be corrected. Some other concerns the authors could address follow.
1. Line 50, what is a "swine chain"?
2. line 49-52, the reader is not sure what this section means
3. single sentences should not be paragraphs.
4. lines 58-61, this is awkward and hard for the reader to understand.
5. line 62, this is not evolution, it is the spread of the strain or replacement of other strains with this one.
6. 66-68, It is uncertain how resistances to metals could lead to resistance to macrophage killing, an explanation, not just a reference is required.
7. 307-310, please rewrite this seciton.
8. 312, types would be better than typologies.
9. 315, what does up to 4C mean? don't you mean down to 4C?
10. 325, happened
11. 331, on own is an awkward phrase.
12. 359 what do you mean by commercialization?
English usage needs to be corrected throughout the manuscript.